# Dissecting FGF Signalling to Target Cellular Crosstalk in Pancreatic Cancer

**DOI:** 10.3390/cells10040847

**Published:** 2021-04-08

**Authors:** Edward P. Carter, Abigail S. Coetzee, Elena Tomas Bort, Qiaoying Wang, Hemant M. Kocher, Richard P. Grose

**Affiliations:** Centre for Tumour Biology, Barts Cancer Institute, Queen Mary University of London, Charterhouse Square, London EC1M 6BQ, UK; e.tomasbort@qmul.ac.uk (E.T.B.); qiaoyingwang@qmul.ac.uk (Q.W.); h.kocher@qmul.ac.uk (H.M.K.); r.p.grose@qmul.ac.uk (R.P.G.)

**Keywords:** pancreatic cancer, FGF signalling, crosstalk, stroma, targeted therapy

## Abstract

Pancreatic ductal adenocarcinoma (PDAC) has a poor prognosis with a 5 year survival rate of less than 8%, and is predicted to become the second leading cause of cancer-related death by 2030. Alongside late detection, which impacts upon surgical treatment, PDAC tumours are challenging to treat due to their desmoplastic stroma and hypovascular nature, which limits the effectiveness of chemotherapy and radiotherapy. Pancreatic stellate cells (PSCs), which form a key part of this stroma, become activated in response to tumour development, entering into cross-talk with cancer cells to induce tumour cell proliferation and invasion, leading to metastatic spread. We and others have shown that Fibroblast Growth Factor Receptor (FGFR) signalling can play a critical role in the interactions between PDAC cells and the tumour microenvironment, but it is clear that the FGFR signalling pathway is not acting in isolation. Here we describe our current understanding of the mechanisms by which FGFR signalling contributes to PDAC progression, focusing on its interaction with other pathways in signalling networks and discussing the therapeutic approaches that are being developed to try and improve prognosis for this terrible disease.

## 1. Introduction

Pancreatic ductal adenocarcinoma (PDAC), the most common and most lethal form of pancreatic cancer is an area of high unmet clinical need [1,2]. One of the world leading causes of cancer-related death, with a five-year survival rate below 8%, there have been no significant advances in diagnosis or treatments over recent years [3,4]. It is predicted to become the second most common cause of cancer related death in the USA by 2030 [5], with an absence of specific symptoms, lack of effective biomarkers and poor response to treatment [6].

PDAC typically manifests within a characteristic, desmoplastic stroma [7], comprising numerous cell types including fibroblasts, immune cells, endothelial cells and pancreatic stellate cells (PSCs), embedded within an abundance of extracellular matrix proteins and polysaccharides, such as collagen, laminin, hyaluronic acid (HA) and fibronectin [8,9,10,11] (Figure 1). This creates a hypoxic environment, limiting the effectiveness of chemotherapy [12] and helping promote tumour development [13].

Understanding the biology of the desmoplasia characteristic of PDAC represents a critical challenge in potentially improving patient outcomes. Activated PSCs are pivotal to the generation of the dense desmoplastic stroma, as well as tumour progression and invasion [7], making these cells an attractive therapeutic target.

## 2. A Role for PSCs in PDAC Invasion

As a key part of the stroma, PSCs make up less than 5% of cells in a healthy pancreas and exist in a quiescent state, but become activated during pancreatic injury and tumour development, begin proliferating and lose their vitamin A stores [7]. Activated PSCs enter into crosstalk with cancer cells to induce tumour cell proliferation and invasion, differentiating into cancer-associated fibroblasts (CAFs) and leading to metastatic spread [10,15,16] (Figure 2).

Analysis of the secretome of activated PSCs highlighted some key proteins that could be involved in re-modelling PDAC stroma to allow tumour invasion [17]. Secretion of ligands such as FGF2, PDGF, TGF-β, CSF-1 and CTGF by activated PSCs has been shown to promote cancer cell proliferation and invasion [18,19,20,21]. Extra-cellular vesicles may act as carriers for many of the growth factors [22]. Reversal of PSC activation, by All-trans retinoic acid (ATRA) or tamoxifen, can decrease actomyosin contractility and, therefore, prevent ECM remodeling and invasion, as well as reducing cancer cell proliferation [15,23,24]. In fact, ATRA has been used in early phase clinical trials in combination with chemotherapy demonstrating its valid use as a stromal targeting agent for patients with pancreatic cancer [25].

Of particular interest, fibroblast growth factor (FGF) signalling has been identified as a key pathway involved in crosstalk between cancer and stellate cells within the desmoplastic stroma of PDAC tumours [10,26,27,28], and will be the focus of this review.

## 3. FGF/FGFR Signalling in PDAC

The FGF signalling pathway is a key regulator of cell growth, development, cell to cell contact and differentiation. As such, it is tightly governed to control cell differentiation and ensure tissue homeostasis, and disruption to FGF signalling has been associated with many different diseases, including cancer (Figure 3; [29]. During PDAC progression, many developmental pathways are activated, including FGF signalling [27]. Cancers driven by FGF signalling can be treated with targeted therapies, such as tyrosine kinase inhibitors (TKIs), as detailed below. This highlights the notion that targeting FGF signalling is a viable clinical option, with therapies already being used in patients. Key to the potential for targeting FGFR signalling in PDAC is that it may well be most effective via targeting the stromal cells rather than the cancer cells themselves. This might circumvent the problems of rapid evolution of drug resistance which plagues TKIs in the clinic, since stromal cells lack the genetic plasticity of cancer cells.

### 3.1. FGF Ligands

In PDAC, FGF signalling is increasingly recognised as one of the critical pathways involved in crosstalk between cancer and stellate cells within the tumour [26]. FGF1 and FGF2 can be overexpressed by cancer cells and have been correlated with advanced tumour stage [30,31]. High levels of FGF2, VEGF and FGF13 have also been correlated with shorter patient survival and increased liver metastasis [32,33]. FGF2 overexpression leads to increased cell proliferation and invasion in cancer cells [34] and antibodies against either FGF2 or FGF receptors causes a 50% reduction in cell proliferation and cell invasion of PDAC cancer cells in rudimentary single cell type 2D cultures, respectively, suggesting FGF signalling could be a good therapeutic target [35,36]. Interestingly, it has also been demonstrated that secretion of FGF2 can lead to increased cancer cell adhesion and inhibition of invasion of cancer cells [37,38,39].

Other FGF ligands, including FGF5, FGF7 and FGF10, have been reported to be overexpressed in PDAC leading to increased cell proliferation and invasion [40,41,42,43]. In particular, FGF7 and FGF10 are reported to signal from the stroma to the cancer cells. FGF7 can activate nuclear factor kappa B (NF-κB) and induce the expression of VEGF, MMP-9 and urokinase-type plasminogen activator [42]. Pre- and post-chemotherapy serum profiles of PDAC patients have demonstrated that FGF10 is differentially expressed in response to gemcitabine and the EGFR inhibitor erlotinib, indicating that changes in FGF10 levels could be used as a predictive biomarker of chemotherapy response in patients [44]. Furthermore, stimulation of PDAC cancer cells with FGF ligands, such as FGF1, FGF2, FGF7 and FGF10, triggers changes in expression of key pancreatic development genes, such as *SRY-related HMG-box gene 9* (*SOX9*), *hepatocyte nuclear factor 3-beta* (*HNF3b*), *hairy and enhancer of split-1* (*HES1*), *GATA-4* and *GATA-6* [45]. Moreover, FGF signalling has been associated with increased expression of inducible nitric oxide synthase (iNOS), indicating that oxidative stress may play a role in FGF-mediated PDAC progression [46]. As well as overexpression of FGF ligands, it has been reported that the levels of FGF-binding proteins (FGF-BP) are increased in PDAC patients [47]. This could indicate a role for FGF signalling mediated re-expression of developmentally important genes in the initiation of PDAC lesions, in accordance with the hypothesis that oncology recapitulates ontogeny [48].

### 3.2. FGF Receptors

FGF receptors, including FGFR1, are also overexpressed within PDAC and can be associated with patient survival [49,50]. Overexpression of FGFR4 has been linked with cancer cell adhesion to ECM proteins, such as laminin and fibronectin, decreasing cell migration and increasing patient survival [51]. Inhibition of FGFR4 can prevent matrix adhesion through neural cell-adhesion molecule (N-CAM), suggesting that this could be a key pathway regulating pancreatic cell migration and invasion [51]. In contrast, FGFR2 overexpression has been linked with shorter patient survival in PDAC [43]. Downregulation or inhibition of FGFR2 can reduce tumour cell survival and migration, as well as tumour development in vivo [52].

Alternative splicing and isoform expression can alter the role of FGF receptors in PDAC. For example, FGFR1α has been linked with decreased cancer cell proliferation and increased anti-tumour responses, whereas expression of FGFR1β can increase resistance to chemotherapy treatment in xenograft models, consistent with an inhibitory role for the first Ig-loop, which is absent in FGFR1β [53]. The third membrane proximal Ig-loop is also important: expression of the IIIc isoform of FGFR2 has been associated with increased cell proliferation and liver metastasis [54]. Contrastingly, expression of the FGFR1 IIIb isoform has been shown to decrease tumour growth in mouse models [55]. Additionally, it has been reported that the overexpression of FGFR1 IIIb in PDAC cells induces the expression of secreted protein acidic and rich in cysteine (SPARC), which regulates cell-cell interactions and decreases tumour progression [56]. FGFR IIIb isoforms are usually expressed on epithelial cells, whilst FGFR IIIc isoforms are expressed by mesenchymal cells [29]. Changes in isoform expression in cancer cells have been linked with tumorigenesis [55]. The expression of FGFR1 isoforms on PDAC cells can be modulated by the stroma, with PSCs inducing cancer cells to switch to the mesenchymal FGFR1 IIIc isoform [41,57].

Using bioinformatic analysis of freely available datasets, we provide a resource for the current understanding of FGF and FGFR expression/alteration in PDAC patients and the most commonly used PDAC cell lines (Figure 4).

## 4. FGFR Signalling Crosstalk

In addition to canonical receptor tyrosine kinase signalling, FGFs/FGFRs can interact with a number of other pathways in pancreatic cancer. Here we consider these interacting partners and their roles in cancer progression.

### 4.1. Wnt/β-Catenin

Wnt and FGF have been shown to act in concert in a number of developmental processes including limb [62], lung [63], somitogenesis [64,65], and pancreatic development [66]. A positive feedback loop exists between FGF and Wnt, in which FGFR activation can promote Wnt ligand expression. Wnt activation of β-Catenin can then promote expression of FGFRs [63], and FGF ligands [67,68].

There are two pools of β-Catenin within cells, a cytoplasmic pool that is constantly turned over until stabilised through Wnt signaling, and a pool present at the adherens junction. FGFs 1 and 2 have been shown to enhance E-Cadherin/β-Catenin complexes at adherens junctions between pancreatic cancer cells [38]. This may have a consequence of controlling the available pool of cytoplasmic β-Catenin, and thus regulating Wnt signalling within cells, either by sequestering β-Catenin away from the cytoplasm [69], or releasing more into the cytoplasmic pool [70]. Wnt- β-Catenin signalling is an important signalling nexus between cancer cells and PSCs [15], as well as having been implicated in resistance to 5-Fluorouracil therapy in PDAC [71].

While FGF-stimulated pancreatic cancer cells exhibited enhanced adhesion and decreased invasion as a result of stronger adherens junctions, evidence from murine models of breast cancer suggests that stronger E-Cadherin may be beneficial for the metastatic potential of cancer cells, by promoting the survival of disseminating cells [72].

### 4.2. Transforming Growth Factor β (TGFβ)

TGFβ has multiple roles in pancreatic cancer effecting both cancer cells and the surrounding tumour microenvironment [73]. One of the primary functions of TGFβ is to promote the activation of PSCs and turn them into cancer-associated fibroblasts, facilitating desmoplasia in PDAC [74]. In addition, activated PSCs can drive the invasion of cancer cells in 3D organotypic models, a process which is dependent on FGF signalling [28]. PSC-produced FGF also stimulates the production of TGFβ from cancer cells [43], thus creating a positive feedback loop in which PSCs promote the invasion of cancer cells, which in turn promote further PSC activation (Figure 2). TGFβ, released from CAFs and mast cells has also been shown to drive resistance to cytotoxic chemotherapy in PDAC, in addition to potentiating immune-suppression [75].

FGF can also be antagonistic to TGFβ. In smooth muscle cells, FGF stimulation causes a decrease in TGF-β signalling, switching the cells from a contractile to a proliferative phenotype [76]. This antagonism is apparent in atherosclerotic plaques, where FGF activation and TGFβ loss is associated with increased disease severity [77].

SMAD4 is a prominent tumour suppressor that is frequently mutated in pancreatic cancer. SMAD4 can be regulated downstream of FGF, Wnt, and TGFβ pathways. FGF signalling can lead to MAPK-dependent phosphorylation of SMAD4, priming it for targeted degradation via GSK3β. Wnt inhibits GSK3β activity, which allows for the potentiation of SMAD4 activation by TGFβ even in the presence of FGF signalling [78]. Two cancer-associated mutations of SMAD4 enhance phosphorylation of SMAD4 by GSK3β, which promotes its degradation and thus loss of function [79].

### 4.3. Sonic Hedgehog (SHH)

SHH is expressed in both tumour cells and the surrounding stroma in PDAC, and exhibits both pro-tumourigenic and suppressive effects [80]. For instance, while SHH inhibition appears to limit metastasis in orthotopic models of pancreatic cancer [81], deletion of stromal SHH accelerates disease [82].

SHH suppresses the early development of the pancreas, and its ectopic expression can disrupt later pancreas morphology [83]. In the endoderm, activin and FGF2 permit pancreas development through the suppression of SHH, and expression of the pancreatic transcription factor PDX1 [84]. However, evidence from limb development and cultures of human embryonic stem cells (hESC) also suggests FGF2 may promote SHH expression [85], and in the case of hESC cultures, suppress pancreatic gene expression [86]. The crosstalk between FGFs and SHH may therefore be context dependent and influenced by companion signalling events. Interestingly, all of the above pathways (WNT, Hedgehog, TGFβ and FGF) can be regulated by the cell surface proteoglycan Glypican-1, suggesting that a common signalling nexus may be a possible therapeutic target [87].

### 4.4. Stemness and Transcriptional Regulation

A subset of cells within a tumour can display characteristics associated with stem cells. These cancer stem cells are often considered resistant to conventional therapy and are responsible for tumour initiation and recurrence [88]. Knockdown or inhibition of either FGFR1 or FGFR2 limits the ability of pancreatic cancer cells to form spheroids, a characteristic of cancer stem cells, suggesting an involvement of FGF signalling in maintaining stemness [54,89,90]. Indeed, inhibition of FGFR reduces the expression of stemness transcription factors Oct4, Nanog, and SOX2 in the pancreatic cancer cell line SUIT-2. FGFR signalling also maintains the survival of a small subset of cells that express ALDH, a stem cell marker [89].

Mechanistically, FGFR signalling maintains the stability of SOX2 in the nucleus via the Akt pathway. Blockade of either FGFR or Akt leads to SOX2 degradation, which can be reversed through ectopic activation of Akt [90]. FGFs may therefore be critical for the maintenance of cancer stem cells within pancreatic tumours.

Mutations in *KRAS* are almost ubiquitous in pancreatic cancer cells, the importance of which is underscored by the prevalence and severity of murine models of pancreatic cancer driven by mutant KRAS [91]. SOX9 has been identified as a key downstream effector of KRAS, which promotes acinar to ductal metaplasia in early pancreatic cancer and accelerates malignancy [92,93].

SOX9 is also found downstream of FGF signalling [94], and is involved in the development of a number of organs including lung [95], lacrimal gland [96], and kidney [97]. SOX9 is also involved in the development of the pancreas, where in addition to promoting ductal cell fate, also promotes FGFR2 expression in progenitor cells, thus fostering a positive feedback loop [26,98].

Myc is overexpressed in many cancers, and its importance in pancreatic cancer has recently been underscored by the observation that Myc activation is sufficient to drive indolent, Kras mutant, pancreatic intraepithelial neoplasm towards PDAC along with the associated stromal changes inherent to PDAC [99]. On the cellular level, stromal derived FGF1 appears to activate Myc expression in PDAC cells through a FGFR-Akt-GSK3β signalling axis. This cellular crosstalk is supported by the strong association between stromal fibroblast content and neoplastic Myc expression [100]. A FGFR3/Myc positive feedback loop has been described in bladder cancer cells [101], suggesting that such a positive feedback loop may also be present in PDAC.

### 4.5. Targeting FGFR in PDAC

Predominantly, agents designed to target receptor tyrosine kinases are small molecule inhibitors that directly target the ATP-binding pocket in the split tyrosine kinase domain of the receptor. For FGFRs these inhibitors fall into two broad categories; multi-tyrosine kinase inhibitors (TKIs) that inhibit multiple kinases in addition to FGFR, and targeted inhibitors that selectively target the FGFR tyrosine kinase [102]. Multi-TKI agents have the most pre-clinical and clinical data, but data on newer, targeted FGFR inhibitors are being reported.

#### 4.5.1. Multi-TKI Drugs

Ponatinib is an inhibitor of BCR-ABL but also targets other kinases, including VEGFR, PDGFR and FGFR, at nM concentrations. Ponatinib appears to be particularly effective in FGFR driven cancers, as demonstrated by an enhanced suppressive effect on the growth of FGFR-driven tumours compared to non-FGFR counterparts in murine models of endometrial, bladder, and gastric cancers [103]. In PDAC xenograft tumours, ponatinib was able to reduce tumour growth, which was enhanced when in combination with a MEK inhibitor [104]. In further support of a use for ponatinib in FGFR driven cancers, a clinical trial using ponatinib to treat patients with advanced biliary cancer that express FGFR2 fusions has recently been run [105].

Dovitinib is another multi-TKI agent that targets FGFR [106,107,108,109]. As with ponatinib, dovitinib was shown to be effective in reducing the growth and metastasis of PDAC orthotopic tumours [110]. In addition, dovitinib is particularly effective in PDAC patient-derived xenografts that harbour enhanced FGFR2 expression [111]. A recent phase 1b study demonstrated effectiveness of dovitinib in combination with gemcitabine and capecitabine in patients with advanced pancreatic cancer [112].

Another multi-TKI agent targeting FGFR, nintedanib, shows efficacy in multiple murine models of PDAC [113,114]. Nintedanib was also able to dramatically reduce tumour growth in the Rip1Tag2 transgenic mouse model of pancreatic neuroendocrine tumours [115]. A phase 1/2 trial of nintedanib in pancreatic cancer is currently ongoing [116].

The multi-TKI agent lenvatinib has also demonstrated effectiveness in PDAC xenografts ectopically overexpressing FGF, as well as xenografts of PDAC cell lines [117]. A phase 2 trial of lenvatinib in pancreatic neuroendocrine tumours has recently completed [118], and a phase 2 trial in various solid cancers including pancreatic cancer in combination with immunotherapy is ongoing [119].

#### 4.5.2. FGFR Selective Inhibitors

More selective FGFR inhibitors have been developed for which pre-clinical data in PDAC have been reported. PD173074 was one of the first selective FGFR inhibitors, and substantially reduced tumour growth and metastasis in orthotopic PDAC tumours and 3D models of PDAC invasion [28,120]. More selective, clinically relevant, FGFR compounds have also been developed, such as AZD4547, which reduces tumour growth and bone metastases of PDAC patient derived xenografts [121]. Another FGFR specific TKI, BGJ398 (Infigratinib), shows effectiveness in reducing tumour growth in PDAC xenografts, particularly in combination with a MEK inhibitor [100]. Clinically, BGJ398 has been shown to be effective in a recent phase 3 trial of cholangiocarcinoma patients who harbour *FGFR2* fusions [122,123].

As with the multi-TKIs, these more specific inhibitors target the same ATP binding pocket, but other methods to target FGF signalling in cancer include ligand traps, FGFR/FGF directed antibodies, and allosteric receptor modulators, each of which are in clinical development [102]. Of relevance to pancreatic cancer is the FGFR allosteric modulator SSR128129E. This compound binds to the extracellular portion of FGFR and prevents the internalisation of the FGF/FGFR complex, which negates intracellular signalling [124]. SSR128129E was able to reduce the growth of orthotopic PDAC tumours, pointing towards a therapeutic role for allosteric inhibition of FGFR [125]. With increasing interest in CAR-T cell therapy, FGFR may act as a potential target in at least a fraction of patients, in a manner similar to recently discovered targets such as CEACAM7 [126].

## 5. Conclusions

There is abundant evidence for FGFR signalling as an actionable target in PDAC. Effectively disrupting FGFR-mediated cross-talk between the tumour and stroma, either alone or in combination with other therapies, could translate to improved therapeutic responses in PDAC patients by providing novel treatment options in the clinic. Intra- and inter-tumoural stromal heterogeneity may define precision stromal targeting of FGFR signalling [127]. By understanding more about how FGFR signalling interacts with other pathways in PDAC, we can develop more rational approaches to combination therapies, with potential to synergise cancer-specific effects whilst avoiding overlapping toxicities, leading ultimately to success in achieving our goal of improved patient outcomes [128].

## Figures and Tables

**Figure 1 cells-10-00847-f001:**
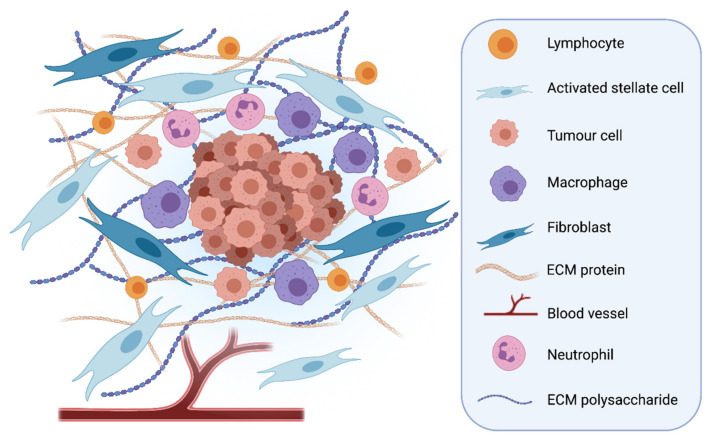
Desmoplastic stroma in Pancreatic Ductal Adenocarcinoma. The stroma makes up a large volume of PDAC tumours. Many cells found within the stroma can contribute to tumour progression and invasion, such as activated pancreatic stellate cells, inflammatory immune cells and cancer associated fibroblasts. The stroma is very stiff due to the high levels of Extracellular Matrix (ECM) proteins that are laid down by activated stellate cells and fibroblasts. PDAC tumours are hypoxic and do not contain many blood vessels [14]. Created with BioRender.com.

**Figure 2 cells-10-00847-f002:**
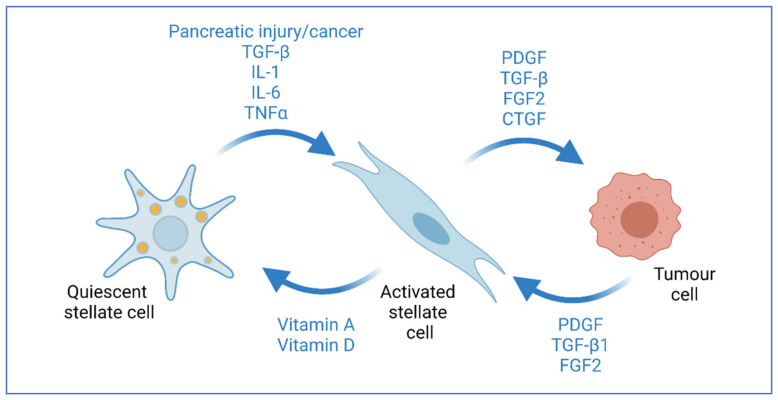
Interaction of stellate cells and cancer cells. Pancreatic stellate cells become activated in response to tissue injury and inflammation. This activation causes the cells to lose their vitamin-filled lipid droplets and express myofibroblast markers. Activated stellate cells in tumours can promote tumour cell proliferation through growth factor secretions, such as FGF2. Tumour cells can also promote stellate cell activation through growth factors, such as TGF-β. Activated stellate cells can be returned to their quiescent state through treatment with All-trans retinoic acid or vitamin D derivatives [7]. Created with BioRender.com.

**Figure 3 cells-10-00847-f003:**
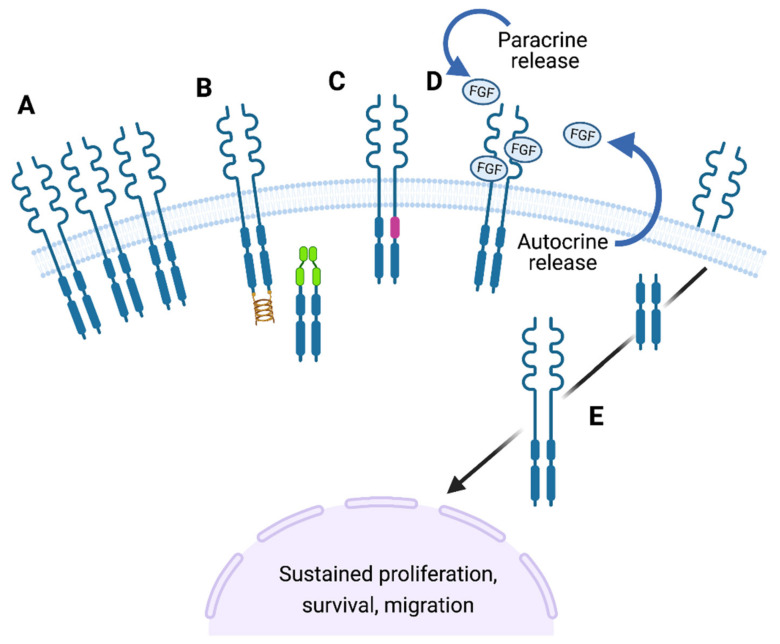
FGF signalling pathway alterations in cancer. The FGF signalling pathway can become altered in a number of methods in cancer to cause sustained cell proliferation and survival: (**A**) Overexpression of FGF receptors cause ligand independent activation, e.g., FGFR2 in gastric cancer. (**B**) Receptor translocations can occur also leading to ligand independent activation of signalling, e.g., FGFR N-terminal fusion with transcription factors (green) or C-terminal fusion with TACC (Transforming Acidic Coiled Coil) proteins (gold). (**C**) Activating mutations (pink) within the FGF receptors cause constitutive activation even in absence of the ligand, e.g., FGFR2 in endometrial cancer. (**D**) Paracrine or autocrine overexpression of FGF ligands can lead to increased FGF signalling, e.g., FGF1 in ovarian cancer. (**E**) Sub-cellular translocation of either the full length or a cleaved version of the receptor can occur, for example to the nucleus where it could act as a transcription factor, e.g., FGFR1 in pancreatic or breast cancer [28,29]. Created with BioRender.com.

**Figure 4 cells-10-00847-f004:**
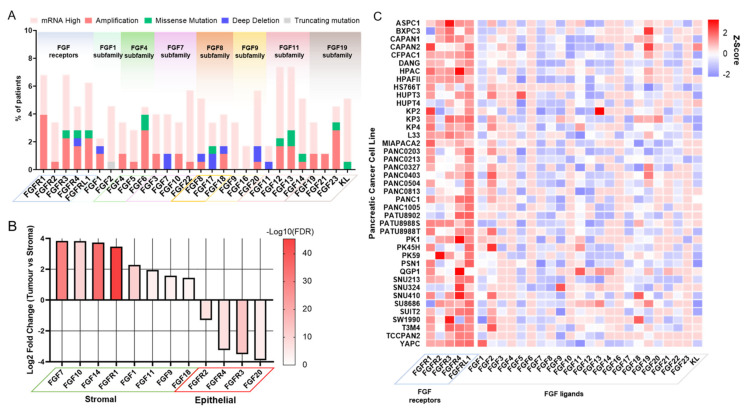
FGF/FGFR expression in PDAC patients and cell lines. (**A**) Mutation and expression data of FGF related genes from pancreatic tumours from TCGA cohort (*n* = 177). Data of mutations and expression were generated using cBioPortal [58], where mRNA high was assigned to a Z-score > 2. (**B**) Significantly enriched FGF related genes from differential analysis results using DESeq2 [59] comparing microdissections of pancreatic tumour epithelial tissue compared to tumour stromal tissue. Positive Log2 fold change indicates higher expression in stromal tissue compared to tumour epithelial tissue and vice versa. The -Log10 transform of the false discovery rate (FDR) is used to show statistical significance. Data were obtained from GSE93326 [60]. (**C**) Gene expression heatmap of FGF related genes in pancreatic cancer cell lines from the Cancer Cell Line Encyclopaedia [61]. The Z-scores were calculated using the mean and standard deviation of all genes.

## Data Availability

N/A.

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
