# Peer review of "Dissecting FGF Signalling to Target Cellular Crosstalk in Pancreatic Cancer"

_cells, 2021, doi:10.3390/cells10040847_

Round 1
Reviewer 1 Report
Overall, this is a comprehensive review that aims to emphasize the importance of FGF pathways in PDAC. The review is well written and thoroughly researched. Some concerns that could (but don't have to) be addressed. First, most of the cited literature on FGF is old and circumstantial, raising the question of how important FGF is for PDACs. Second, although the review is very comprehensive, it lacks synthesis of ideas. Much information is given but it would help tremendously if a common thread emerged. Thus, moving from FGF to TGFb to Hedgehog to Wnt to sox9 muddles the question, are FGF pathways really important for PDACs and, if so, which and how. The authors could expand their therapeutic targeting part and explain better why the mechanism of action of the various drugs specifically connects the drugs' efficacy to FGF signaling. Most examples are multi-TKI agents that also target some (or all?) FGFRs. Perhaps a few sentences that connect threads or ideas would help improve the significance of this review.
Minor comment:
The idea that "SMAD4 brings together signalling from FGF, Wnt, and TGFβ pathways in its regulation" doesn't make any sense.
Author Response
We thank the reviewer for their kind and thoughtful comments. We appreciate the point about some disconnection in the original manuscript. We have sought to improve this by keeping signalling pathways together and providing some unification in with the links to common extracellular mediators (eg Glypican-1) (lines 330-2) and shared downstream nexus (SMAD4), and then following that with a single stem cell focussed subsection without splitting into individual transcription factors (line 335).
Likewise, we have taken on board their comments on the therapeutic section - clarifying mechanism of action and keeping the multi-TKIs in one section to aid clarity. We have clarified our targeting section (line 371) adding mechanisms (line 373, 429) and splitting into mTKI (line 390) and FGFR specific (line 417).
We thank them for their comments on TGF beta signalling, which indeed were not clear and have been rectified. (lines 302-5)
We have also set the scene with respect to stromal targeting (lines 149-153) to improve the clarity of our message.
These changes have improved the flow significantly and we are grateful for the reviewer's help in this.
Reviewer 2 Report
Comments to the author
In the present manuscript, Carter Ed P. et al. demonstrated that the importance of FGF signaling and their cellular crosstalk in pancreatic cancer.
Pancreatic ductal adenocarcinoma (PDAC) has a poor prognosis and is challenging to treat due to their desmoplastic stroma and hypo-vasculature, which limits the effectiveness of chemotherapy and radiotherapy. Therefore, identification of the FGF signaling and their cellular crosstalk in PDAC is very valuable study for discovery of therapeutic strategies.
Major concerns:
- In the 2. A role for PSCs in PDAC invasion, authors presented only PSC as a source of FGF, which can activate FGF signaling in PDAC. Actually, it is well known that cancer-associated fibroblasts (CAFs) can also express and secret FGF. Although, in line 218, authors described ‘One of the primary functions of TGFb is to promote the activation of PSCs and turn them into cancer-associated fibroblasts, facilitating desmoplasia in PDAC’, since authors show both PSC and CAF in Figure 1, and do not describe the two cells separately in the manuscript, it is need to explain CAFs as the source of FGF. Also, title 2 ‘A role for PSCs I PDAC invasion’ needs to be modified according to the content.
- In the paragraph 4, FGFR signaling crosstalk is described. However, in the abstract part, it was described that the major signaling pathway related to FGF in PSC of PDAC are chemoresistance or radiotherapy resistance. Therefore, in order to make a review paper with more structured and rich content, it is needed to add a description and reference paper on the resistance to chemotherapy and radiotherapy for each gene. For example, Stewart DJ. Wnt signaling pathway in non-small cell lung cancer. J Jatl Cancer Inst. 2014 Jan;106(1):djt356, Cui J. et al., Role of Wnt/b-catenin signaling in drug resistance of pancreatic cancer. Curr Pharm Des. 2012;18(17):2464-71, Kurimoto R et al., Drug resistance originating from a TGF-β/FGF-2-driven epithelial-to-mesenchymal transition and its reversion in human lung adenocarcinoma cell lines harboring an EGFR mutation. Int J Oncol. 2016 May;48(5):1825-36, Birous-Leprieur E. et al., Hedgehog Signaling in lung cancer: from oncogenesis to cancer treatment resistance. IntJ Mol Sci. 2018 Sep 19;19(9):2835, Yang H et al., Extracellular ATP promotes breast cancer invasion and chemoresistance via SOX9 signaling. Oncogene. 2020 Aug;39(35):5795-5810.
Minor concerns:
- In the Abstract part, please describe ‘FGFR’ in full name, and indicate with abbreviation.
- Line 38, what means ‘this’?
- In the right small box of Figure 1, it is needed to increase picture size of both the ECM protein and ECM polysaccharide.
- In the legend of Figure 1, please describe ‘PDAC’ and ‘ECM’ in full name, and indicate with abbreviations.
- In the legend of Figure 2, please describe ‘PDAC’ and ‘ECM’ in full name, and indicate with abbreviations.
- Line 77, author described ‘therefore, prevent …., as well as reducing cancer cell proliferation’. That means ‘prevent reducing cancer cell proliferation.’?
- In the Figure 3, please express the information described in the legend of Figure 3.
- In the legend of Figure 3, please describe ‘FGF’, ‘FGFR’, and ‘TACC’ in full name, and indicate with abbreviations.
- In the part of 3.1, it is needed that a description and reference about the effects of FGF1 on characteristics of cancer. For example, Clinicopathological significance of fibroblast growth factor 1 in non-small cell lung cancer (Li J. et al., Human pathology. 2015 Dec;46(12):1821-8).
- At the end of paragraph 3.1, a total of 5 genes and 2 proteins are described. Moreover, the conclusion of this section was ‘initiation of precursor PDAC lesions’. However, the explanation of these factors does not mean ‘initiation of precursor PDAC lesions’. So, to deduce these conclusions, additional explanations of the gene functions involved in the conclusions are needed.
- It is needed to compose the contents of are FGF and FGFR targeted therapy, and combination therapy with targeting aforementioned gene in the paragraph 4.7.
Author Response
We thank the reviewer for their detailed and constructive comments on our manuscript. We have made significant changes as a result, including improving figures 1 (incorporating changes to the key and the legend) and 3 (incorporating addition of two separate versions of FGFR fusions in B - representing the two key mechanisms - and also improving the legend for clarity.
We realise that we had not made clear that pancreatic stellate cells themselves are a major source of cancer associated fibroblasts - hence the lack of clarity they had identified in their comments about sources of FGF. We have now added text to make this very clear (line 110)
We have also added full text for abbreviations throughout as requested (lines 15, 90, 93, 118, 147, 159, 160.
We have added references to drug resistance, focusing on PDAC rather than lung cancer (line 276, 293-295)
We have added the reference to FGF1 clinicopathologic significance and clarified our comments on the significance of developmental genes in PDAC (line204-5)
We have clarified our targeting section (line 371) adding mechanisms (line 373, 429) and splitting into mTKI (line 390) and FGFR specific (line 417)
We have made further changes to improve clarity by including a section on discussing the benefits of targeting the stroma specifically (lines 149-53).
We thank the reviewer for the guidance to help us improve our submission and trust that they approve of the changes.
Reviewer 3 Report
The aim of the work was a critical review of the literature on Fgf-induced signaling in modulating cellular crosstalk in adenocarcinoma.
This reviewer believes that a hasty review of the literature has been carried out with little attention to recent publications on adenocarcinoma / FGF /signaling / kinase inhibitors .
This paper needs a thorough review and an update
Author Response
It is very difficult to respond to such a curt and unsubstantiated review. Our review is specifically addressing PDAC as opposed to adenocarcinomas in general, as the reviewer states, and the idea that this is a "hasty review" is, quite frankly, offensive. We thank the other reviewers for their constructive criticism but request that this reviewer either give some detail to substantiate their claims or is ignored.
Round 2
Reviewer 2 Report
Dear authors,
I hope you are doing well.
In the present manuscript, Carter Ed P. et al. demonstrated that the importance of FGF signaling and their cellular crosstalk in pancreatic cancer.
As a result of reviewing the revision of your manuscript, authors responded with great sincerity to the concerns presented by the reviewer.
I hope you have a good result.
Thank you for your time and efforts,
With warm regards.